# Performance Optimization of Ultralow-Frequency Electromagnetic Energy Harvester Driven by Eccentric mass

**Jintao Liang** *, **Chao Zhang** and **Kangqi Fan**

School of Mechano-Electronic Engineering, Xidian University, Xi'an 710071, China; zhangchao@xidian.edu.cn (C.Z.); kangqifan@gmail.com (K.F.)
* Correspondence: jtliang@xidian.edu.cn; Tel.: +86-137-7210-4018

**Abstract:** Driven by an eccentric mass through a two-layered cantilevered plectrum, the electromagnetic energy harvester (EEH) can convert low-frequency mechanical vibrations into continuous uni-directional rotation. To optimize the performance of the EEH, electromagnetic analysis of the EEH was conducted. Three-phase winding permanent magnet (PM) topology was employed, and combinations of different coils and magnet pole numbers were designed. Then, the finite element method (FEM) was applied to analyze the influence of the combinations of the coils and pole numbers as well as the PM dimensions on the three-phase induced voltage. Prototypes with different configurations were fabricated and the analysis effectiveness was confirmed. Furthermore, different types of stator yokes were designed to enhance the magnetic field. Compared to the original prototype, the output voltage of the optimal prototype increased by 0.5 V with the same rotation speed, and the harmonic components were sufficiently low. Then, experiments with excitation by linear reciprocating motions and swing motions were conducted. Under different exciting conditions, the optimal prototype can also induce the highest voltage amplitude. With an increase in the weight of the eccentric mass, a long duration can be reached that lasts up to 12 s. In summary, the proposed optimization can achieve a high-efficiency and high-power density EEH.

**Keywords:** electromagnetic energy harvester (EEH); three-phase machine; finite element method (FEM); harmonic analysis; ultralow-frequency vibration

## 1. Introduction

Harvesting energy from the surrounding environment and transforming it into electric energy is a feasible process that can supply power to wearable, portable and implantable devices, including wireless miniature sensors, etc. [1–4]. Low- and ultralow frequency vibrations widely exist within the natural environment and are considered to be an attractive sustainable resource for replacing traditional chemical batteries. Mechanical vibration contains various kinds of linear and swing vibrations. However, how to capture these mechanical energies efficiently is a great challenge, especially regarding ultralow frequency vibrations [5,6]. The main principles of mechanical energy harvesting include the piezoelectric effect [7,8], electromagnetic effect [9,10], electrostatic effect [11,12], triboelectric effect [13,14], etc. By contrast, electromagnetic harvesters exhibit higher power and longer service time.

Many research studies have been conducted to design various electromagnetic energy harvesters (EEH) for low- and ultralow-frequency vibrations. For instance, Halim et al. [15] proposed a non-resonant EEH, where a free non-magnetic ball in a cylinder periodically hits two magnets suspended on two springs at either ends of the cylinder, and the voltage is induced in the coil wrapped around the cylinder. The prototype generated 110 μW of average power with a 15.4 μWcm$^{-3}$ average power density. Gutierrez et al. [16] designed a 2D magnetic levitation kinetic energy harvester, where a disk magnet lies on a 2D plane, the boundary of which is constrained by a circular sidewall, and cuboidal magnets are distributed around the sidewall to provide a spring force. The prototype obtained a resonant

frequency of 8.2 Hz and an output power of 41.0 μW and 101 μW at 0.1 g and 0.2 g, respectively. Zhao et al. [17] explored an EEH with a magnetic orbit (EMH-MO) that has a circular uniform and low potential energy orbit, and where the moving magnet is magnetically modulated to move regularly along the magnetic orbit under arbitrary excitation. Under a reciprocating motion of 10 Hz and 20 mm, the open-circuit peak-peak voltage was 4.3 V, and the average power was 0.33 mW. Zhang et al. [18] presented a rotational EEH that comprises a twist driving rod and a ratchet-clutch. When the twist rod is compressed, the ratchet can keep rotating inertially for about 20 s. An overall energy of 85.2 mJ can be captured, and a peak power output of 32.2 mW was achieved. Although the latter two rotational EEHs can increase the level of electric power prominently, a relatively large vibration amplitude is generally required to make these harvesters operate efficiently.

For swing motion harvesting, an eccentric mass or pendulum structure is needed to scavenge for the swing vibrational energies [19]. For example, Smilek et al. [20] presented a Tusi couple energy harvester based on a proof mass with permanent magnets rolling in a circular cavity, where an average power of 5.1 mW from a 1.3 g periodic acceleration waveform at 2.78 Hz can be obtained. Hou et al. [21] introduced a rotational pendulum triboelectric-electromagnetic hybrid generator. With the rotation of the magnet pendulum, the four coils induce the electromagnetic power, and the Cu ring around magnets periodically contacts with the FEP blades to generate the triboelectric induced power. Through the two modes, the maximum power density of 3.25 W/m$^2$ and 79.9 W/m$^2$ was obtained at a vibration amplitude of 14 cm and frequency of 2 Hz. Li et al. [22] designed an eccentric pendulum-based EEH, where the eccentric rotor is arranged with permanent magnets and two sets of coils are symmetrically distributed on both axial sides of the eccentric rotor. The center working frequency is 2.0 Hz with an average output power of 8.37 mW. Halim et al. [23] employed a torsional spring to enhance the harvesting capacity of the eccentric magnetic rotor. The optimal prototype obtained a maximum 61.3 μW average power under a pseudo-walking motion of 1 Hz and ±25° amplitude. Liu et al. [24] developed a non-resonant rotational EEH with a cylindrical stator and a disk-shaped rotor. The rotor magnet can easily rotate around the stator due to external vibrations, and four wound coils arranged around the stator magnet can induce the electrical voltage. Under an 8 Hz excitation, the EEH can supply a maximum power of 10.4 mW at a load resistance of 100 Ω. Since the above eccentric masses or pendulums are excited to swing at small angle and low speed, these harvesters generally produce low electric power and voltage.

In order to capture low- and ultralow- frequency vibrations efficiently, an innovative eccentric mass combined with a two-layered cantilevered plectrum was proposed in our previous study [25]. However, the waveform and the efficiency of the electromagnetic induction effect was not considered and optimized. Therefore, in this study analysis, an optimization design of the electromagnetic performance is carried out to obtain a high efficiency, high power density, and low harmonic components of the inducted voltage. Firstly, a three-phase PM topology with fractional-slot concentrated-winding stator is employed, which can generate three-phase symmetric sinusoidal voltage. Next, how to design the combinations of coils and PM pole numbers is discussed. Then, to enhance the voltage amplitude and reduce the harmonic components, finite element method (FEM) is conducted, and the influence of the coil-pole combinations, the dimensions of PMs, and the types of stator yoke are analyzed. The corresponding prototypes are manufactured and tested, resulting in numerical FEM results that show good agreement with the experimental results. Moreover, the prototypes are installed to a crank-link sliding mechanism for linear reciprocating excitation and a crank-link swaging mechanism for swing motion. Different frequencies and amplitudes of vibrations are loaded, and the induced voltage waveforms are compared.

## 2. Design of EEH

### 2.1. Vibrational Exciting Principle

The structure of the proposed EEH is shown in Figure 1. The eccentric mass can be excited by external mechanical vibrations and transfer the kinetic energy to the rotor via a two-layered cantilevered plectrum that pushes the ratchet set at the inner surface of the rotor. The two-layered cantilevered plectrum is inserted into the central arc of the eccentric mass, which includes a left elastic layer and a right rigid layer. The effective length of the elastic layer (from its clamped end to the tip) is long enough to touch the tooth of the ratchet, whereas the rigid layer is a little shorter, avoiding connection with the tooth. As shown in Figure 2, when the eccentric mass is excited and spins clockwise, the cantilevered plectrum exhibits low stiffness and low frictional resistance against the rotor. On the contrary, when the eccentric mass is excited and spins anticlockwise, the cantilevered plectrum exhibits sufficient stiffness to drive the rotor homodromy. Thus, the rotor can be driven and accelerated to a high unidirectional speed by the eccentric mass, even though the excitation is ultralow-frequency, irregular or intermittent.

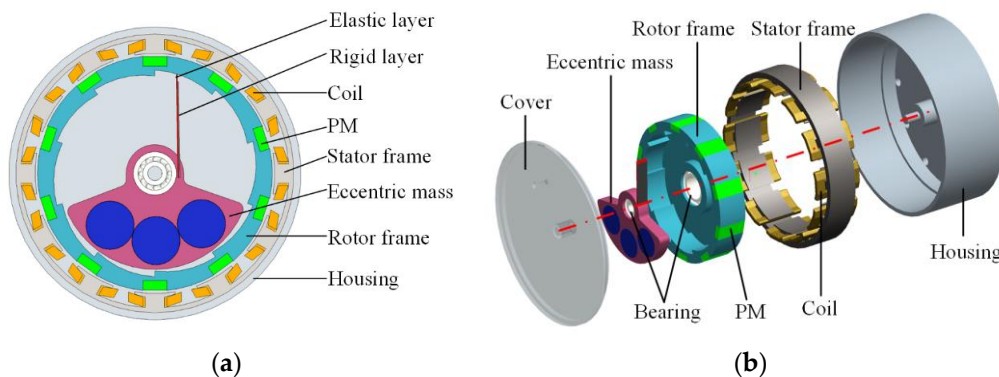

(**a**)         (**b**)

**Figure 1.** Structure of the proposed EEH: (**a**) cross-section view; (**b**) Exploded view.

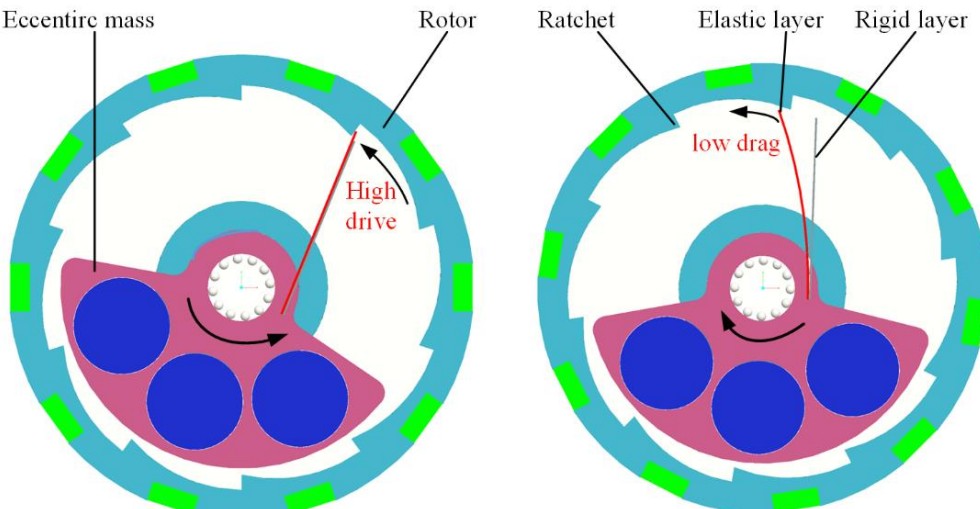

**Figure 2.** Principle of the proposed EEH.

### 2.2. Stator and Rotor Topology

As shown in Figure 1, the outer surface of the rotor frame is uniformly embedded with a sequence of rectangular PMs that are radially magnetized. The polarity of the adjacent two PMs is the opposite. The stator frame has a circumferential teeth-slot structure, in which three-phase individual concentrated coils are wound around a tooth in sequence (A, B, C), and each phase winding consists of several concentrated coils in a series (A1-A2-A3-A4, B1-B2-B3-B4, and C1-C2-C3-C4). When the rotor spins, the rotational

magnetic field is established, and then the three-phase sinusoidal voltage can be in-
duced in the three-phase winding and conveniently transferred to the DC voltage through
a three-phase rectifier bridge. For this three-phase PM topology, the number of coils $Q$
must be a multiple of 3 (three phase); an even number can avoid the magnetic unbalance.
Thus, $Q$ can be set to 6, 12, 18, etc. When $Q$ is less than 12, the leakage flux would become
more, and when $Q$ more than 18, the structure would become complicated. As such, the
preferable number of $Q$ is 12 or 18. Correspondingly, the number of PM poles $p$ should
be an even number for the alternant polarity. When the rotor spins across two adjacent
PMs, it represents one electrical period (360° electrical angle). Therefore, the electrical angle
between adjacent coils can be expressed as:

$$\alpha_c = \frac{180° p}{Q} \tag{1}$$

The number of coils per pole and per phase can be expressed as:

$$q = \frac{Q}{pm} = \frac{z}{d} \tag{2}$$

where $m$ is the number of phases. When $q$ is a fraction, it is called fractional-slot winding.
$q = z/d$ represents an irreducible fraction, which means one phase has $z$ coils in $d$ poles.

As such, each coil can use a 2D vector to indicate its position and magnitude, and the
coil vectors in the same phase can be combined as the vector of one-phase winding. Then,
three-phase vectors are mutually separated at a 120° electrical angle, which can be satisfied
by configuring the number of $Q$ and $p$ properly. For instance, if $p$ is set to 10 and $Q$ is set to
12, then the adjacent coil vectors are staggered where $\alpha_c = 150°$ electrical angle. The coil
and phase vectors are shown in Figure 3a. The No. 1 and 2 coils are combined as vector
A+, while No. 7 and 8 coils are combined as vector A−. If the No. 7 and 8 coils are wound
reversely, then the direction of vector A− becomes the same as vector A+, combining
together as the phase A vector. This is the same for the phase B and Phase C vectors, which
are staggered at a 120° electrical angle respectively.

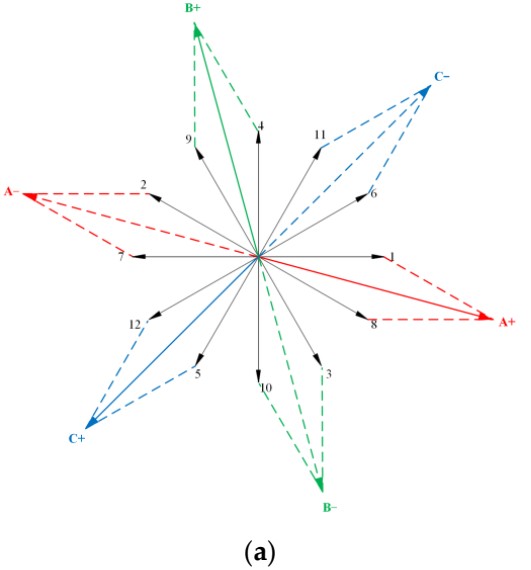

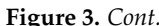

(**a**)

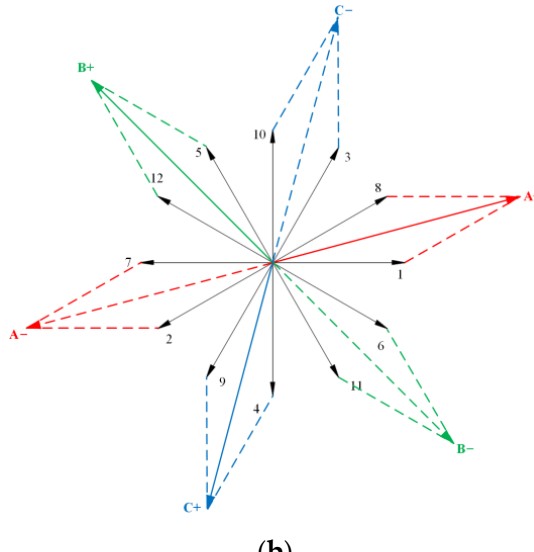

(**b**)

**Figure 3.** *Cont.*

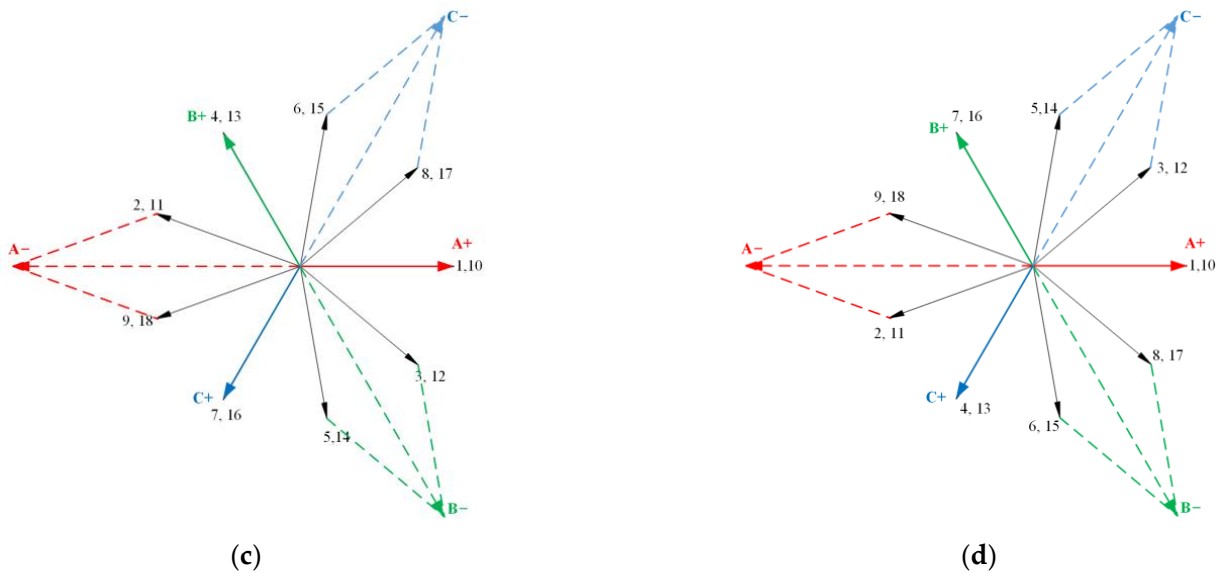

**Figure 3.** Coil and phase vectors with different configurations: (**a**) 12C/10P; (**b**) 12C/14P; (**c**) 18C/16P; (**d**) 18C/20P.

Because the coil vectors of the same phase are not in the same direction, the winding factor should be considered to enhance the output voltage. The winding factor $k_w$ is defined as the product of the winding distribution factor ($k_d$) and the short pitch factor ($k_c$). The distribution factor measured the resultant voltage of the distributed winding regards concentrate winding. $k_d$ can be expressed as:

$$k_d = \frac{\sin(\frac{\pi}{2m})}{z \sin(\frac{\pi}{2mz})} \tag{3}$$

The short pitch factor $k_c$ is the measure of the number of armature slots between the two sides of a coil. $k_c$ can be expressed as:

$$k_c = \sin(\frac{\alpha_c}{\pi} \frac{\pi}{2}) = \sin(\frac{\alpha_c}{2}) \tag{4}$$

Thus, the winding factor $k_w$ is:

$$k_w = k_d k_c \tag{5}$$

A higher winding factor can induce larger voltage amplitude. Initial 12C/10P ($k_w = 0.933$), 12C/14P ($k_w = 0.933$), 18C/16P ($k_w = 0.945$) and 18C/20P ($k_w = 0.945$) are also suitable for configuration, following the phase vector arrangement as shown in Figure 3b–d.

### 2.3. Dimensions and Materials

In order to reduce the weight of the EEH for a wearable device, the structure components, including the eccentric mass frame, rotor frame and stator frame, were fabricated using a resin material via a 3D printer. To enhance the magnetic flux and generate a larger voltage, the yoke section of the stator frame can be produced using a permeability magnetic material, such as a laminated silicon steel sheet. The following electromagnetic analysis was used to compare this difference. For manufacturing the experimental prototype, the outer and inner diameters of the stator were set to 60 mm and 50 mm, respectively, and the outer diameter of the rotor was set to 49 mm. Initially a 12/10 coil-pole configuration was selected. The dimensions of each PM were 10 mm × 5 mm × 2 mm, and the number of turns per coil was 60.

The fan-shaped resin frame of the eccentric mass with a central angle of 130° and radius of 20 mm was embedded with three identical cylindrical copper blocks (thickness: 4 mm, diameter: 10 mm). For comparison, the eccentric mass can be fabricated as a whole block of steel to increase its rotational inertia. The rigid layer of the cantilevered plectrum with a size of $18.5 \times 4 \times 0.2$ mm$^3$ was made by a copper sheet, while the elastic layer was cut from polyethylene terephthalate (PET) film that has a low friction coefficient of 0.04 to form a size of $19.5 \times 4 \times 0.2$ mm$^3$.

The exploded view of the whole EEH is shown in Figure 1b. The stator was embedded in the housing, which includes a step shaft at the rear central. The eccentric mass and the rotor were connected to the shaft through two different ceramic bearings. A cover was attached to the front of the housing to prevent air resistance. Then, the initial prototype was implemented, exhibiting a 63 mm outer diameter, 22.5 mm axial length and 67.6 g weight, as shown in Figure 4.

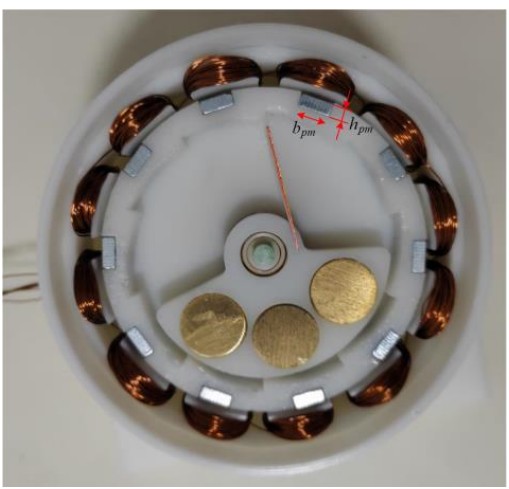

**Figure 4.** Initial prototype of the proposed EEH.

## 3. Electromagnetic Analysis

### 3.1. Comparison of FEM and Experimental Results

The electromagnetic performance of the proposed EEH can be simulated by FEM through the opensource software FEMM, MaxFem, and the commercial software Anasys, Comsol, etc. The general processes are approximately the same for each of these software packages. The initial prototype, including stator and rotor topology, can be established as the 2D electromagnetic geometric model for its cross-section, as shown in Figure 5a. The material property of each part is defined, and excitation and boundary conditions are set. Then, the finite element mesh is generated, as shown in Figure 5b. According to the following partial differential equation of magnetic potential vector $A_z$, numerical computation can be conducted by the FEM solver.

$$\nabla \times \frac{1}{\mu} \nabla \times A_z = -\sigma \dot{A} + J_z \tag{6}$$

where $\mu$ is the permeability, $\sigma$ is the conductivity, $J_c$ is the current density and $\nabla$ represents the Hamiliton operator. On this basis, the other parameters, including magnetic density $B$, magnetic flux $\Phi$, induced voltage V, etc., can be obtained through postprocessing.

For instance, the rotor can be simulated to spin for an electrical period with different speeds, which enables the induced three-phase voltage waveforms to be obtained. As shown in Figure 6a, the three-phase voltage waveforms are obtained when the rotor speed is set to 600 rpm. To verify the accuracy of the FEM results, the initial prototype was connected to a brushless DC motor, which can be regulated to drive the prototype at different speeds, as shown in Figure 7. The experimental results of the induced voltage waveforms at

600 rpm were obtained, as shown in Figure 6b. By comparison, in one period of Phase A (Figure 6c), the experimental results are is in a good agreement with the predicted FEM results in regard to both the waveform and peak values.

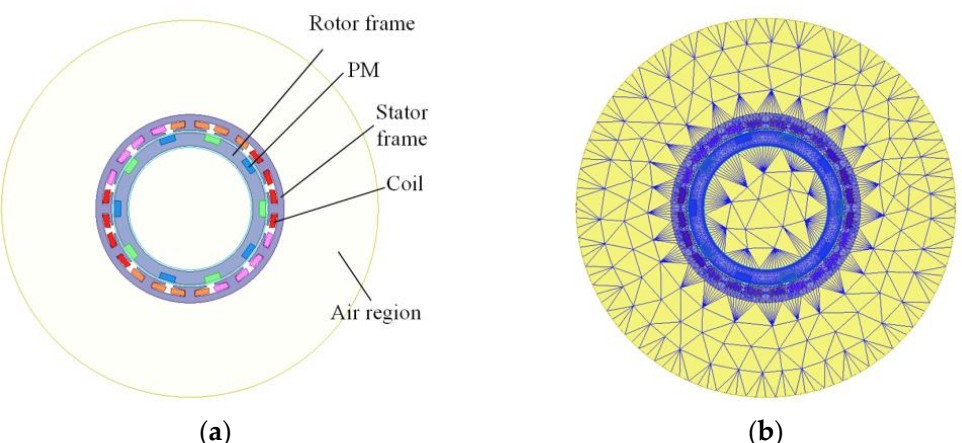

(**a**)　　　　　　　　　　　　　　　　　　　　(**b**)

**Figure 5.** Electromagnetic FEM model of the EEH: (**a**) geometric model; (**b**) mesh generation.

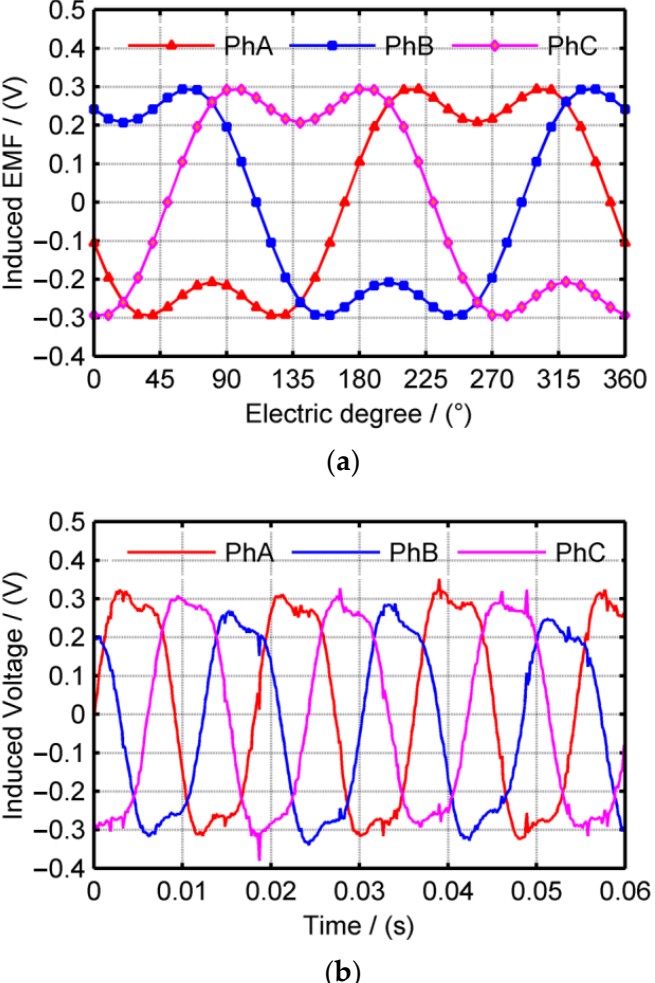

(**a**)

(**b**)

**Figure 6.** *Cont.*

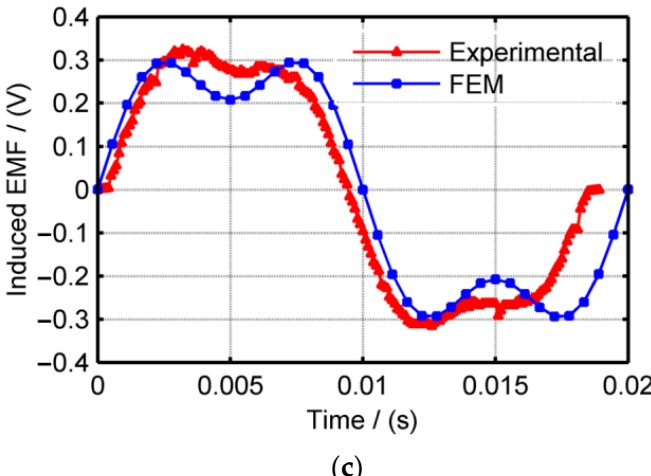

(**c**)

**Figure 6.** Three-phase voltage waveforms with initial model at 600 rpm: (**a**) FEM result; (**b**) Experimental result: (**c**) Result comparison in one period of Phase A.

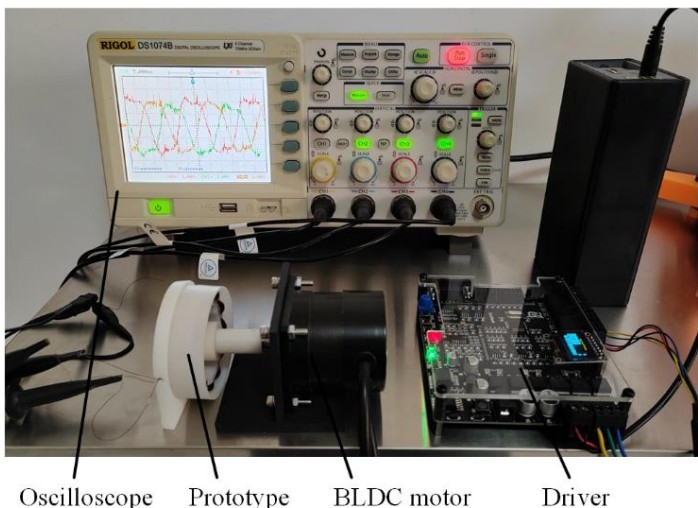

Oscilloscope    Prototype    BLDC motor    Driver

**Figure 7.** Experiment device for voltage measurement.

Moreover, the induced voltage waveforms have excessive high harmonic components that should be reduced. Therefore, the influences of the design parameters were analyzed through FEM.

*3.2. Magnetic Field Optimization*

The dimensions of the PM affect the intensity of the magnetic field directly. The axial length is equal to a rotor length of 10 mm. To reduce the magnetic resistance, the thickness of the PM $h_{pm}$ should be set to a small value; this value was determined to be 2 mm, as shown in Figure 4. Thus, the width of the PM $b_{pm}$ is regulated at different values of 5 mm (initial), 6 mm, 7 mm, 8 mm and 9 mm. The corresponding results of the induced voltage waveforms are shown in Figure 8a. To present a distinct display, only the phase-A waveforms are compared.

With the increase in $b_{pm}$, the voltage amplitude increases, and exhibits a more fundamental sine. Harmonic analysis is shown in Figure 8b. The main harmonic component is the three-time harmonic, when $b_{pm}$ = 5 mm, and the total harmonic distortion (THD) is about 34%; when $b_{pm}$ = 9 mm, the THD reduces to only 2.0%.

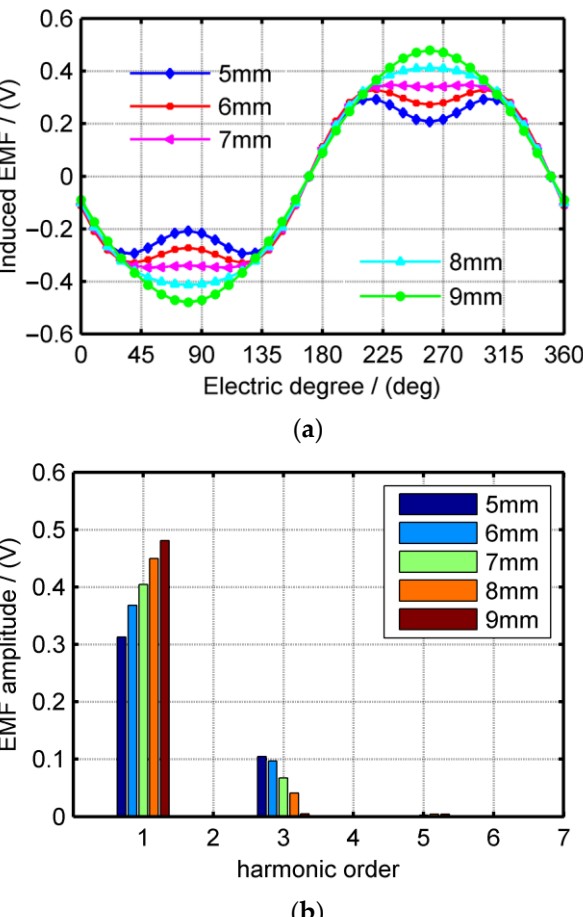

**(a)**

**(b)**

**Figure 8.** FEM results of initial 12C-10P model with different $b_{pm}$: (**a**) Voltage waveforms; (**b**) Harmonic magnitude of voltage.

Employing a permeability magnetic material to produce the stator core can enhance magnetic density and reduce the flux leakage. The stator core is composed of the teeth and the yoke. The yoke is a magnetic frame between teeth that can pass the magnetic flux, and also houses the coil windings. However, if the whole stator is laminated by silicon steel sheet, cogging torque generated between the stator teeth and PM develops tremendous spinning resistance. Thus, a non-teeth stator core is designed, including a resin skeleton and a yoke laminated by silicon steel sheets, as shown in Figure 9. Compared with the initial FEM model, the voltage amplitude increases by about 65%, as shown in Figure 10.

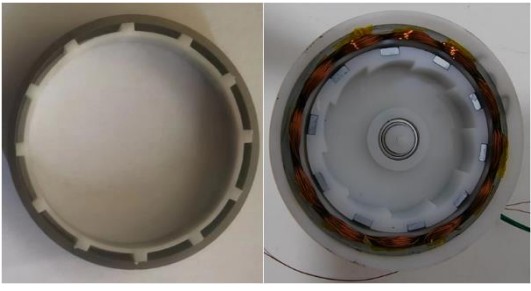

**Figure 9.** Non-teeth stator core structure.

### 3.3. Configuration of Coil and Pole Number

Based on the advanced stator core, the influence of different coil-pole configurations was analyzed. Figure 11 shows the voltage performance of the 12C-14P FEM model. Compared with the 12C-10P model, the voltage waveforms with different $b_{pm}$ have favorable

fundamental sinusoidal voltages, and the harmonic components are sufficiently low. Moreover, the values of voltage amplitude are larger than the 12C-10P. When $b_{pm}$ = 6 mm with 14 poles (10 mm × 6 mm × 2 mm × 14 × 7500 kg/m$^3$ = 0.0126 Kg), the amplitude is about 0.95 V. By contrast, when $b_{pm}$ = 9 mm with 10 poles (10 mm × 9 mm × 2 mm × 10 × 7500 kg/m$^3$ = 0.0135 Kg), the amount of PMs is larger, but the amplitude is only 0.80 V.

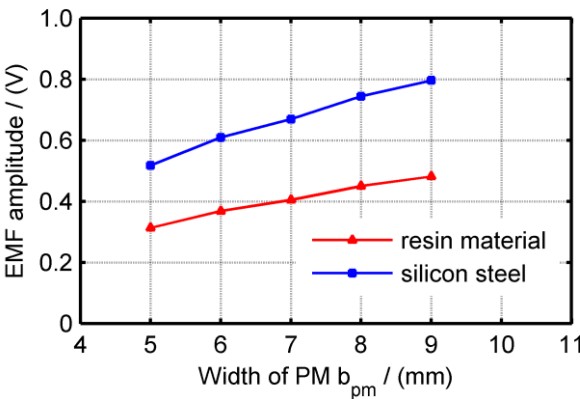

**Figure 10.** Voltage amplitude of 12C-10P FEM model with different material.

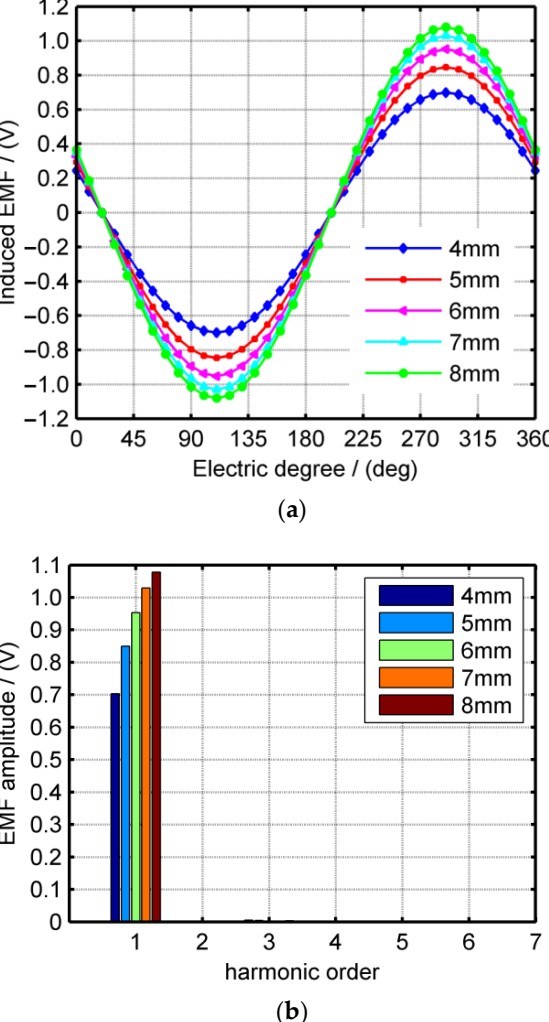

**(a)**

**(b)**

**Figure 11.** FEM results of steel-yoke 12C-14P model with different $b_{pm}$: (**a**) Voltage waveforms; (**b**) Harmonic magnitude of voltage.

The 18C-16P and 18C-20P configurations were also analyzedand the voltage amplitude results of different configurations were compared, as shown in Figure 12a. In addition, to indicate the magnetic efficiency of the PM, the ratio of the voltage amplitude versus the PM mass was also compared, as shown in Figure 12b. Moreover, the values of voltage amplitude, harmonic THD and voltage per PM mass with different configuration are listed for comparison in Table 1.

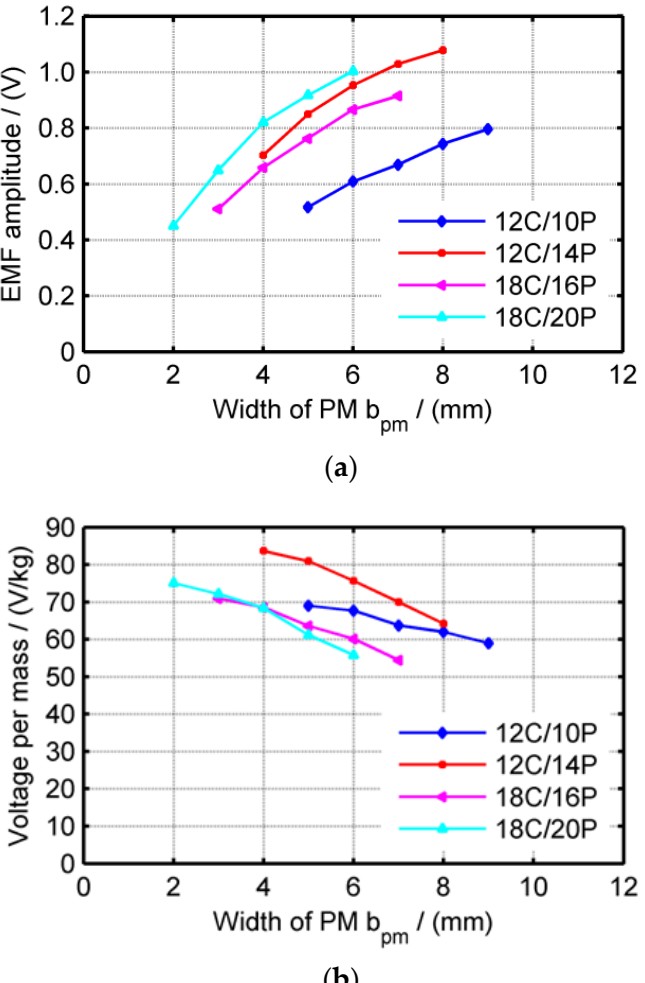

**Figure 12.** Comparison of voltage performance with different configurations: (**a**) Voltage amplitude; (**b**) Voltage amplitude versus PM mass.

In general, the 12C-10P and 18C-16P configurations exhibit a high voltage hamonic component. By contrast, the harmonic THD of 12C-14P and 18C-20P are sufficiently low, all less than 1%. With the increase of $b_{pm}$, the voltage amplitude values of every configuration would increase uniformly, but the magnetic efficiency of every configuration would decrease. This is because although the 12C-14P and 18C-20P configurations exhibit a larger value (over 1.0 V), 18C-20P has a low ratio of voltage amplitude to PM mass, while the ratio of 12C-14P is the highest. This is due to the magnetic resistance of the non-magnetizing resin material being large, and the flux leakage increasing when the PM intensity increases. To reduce the cost and weight, 12C-14P with $b_{pm}$ = 6 mm is determined as the optimal configuration, as shown in Figure 13. Meanwhile, 18C-20P with $b_{pm}$ = 4 mm was also manufactured to verify the effectiveness of the FEM results.

**Table 1.** Comparison of voltage values with the different configurations and $b_{pm}$.

| Configurations | Voltage Amplitude | Harmonic THD (%) | Voltage per Mass (V/Kg) |
|---|---|---|---|
| 12C10P $b_{pm}$5 mm | 0.517 | 26.86 | 68.997 |
| 12C10P $b_{pm}$6 mm | 0.609 | 21.09 | 67.684 |
| 12C10P $b_{pm}$7 mm | 0.669 | 13.74 | 63.739 |
| 12C10P $b_{pm}$8 mm | 0.744 | 7.81 | 61.983 |
| 12C10P $b_{pm}$9 mm | 0.796 | 1.44 | 58.971 |
| 12C14P $b_{pm}$4 mm | 0.703 | 0.74 | 83.682 |
| 12C14P $b_{pm}$5 mm | 0.850 | 0.56 | 80.923 |
| 12C14P $b_{pm}$6 mm | 0.953 | 0.29 | 75.611 |
| 12C14P $b_{pm}$7 mm | 1.029 | 0.21 | 70.002 |
| 12C14P $b_{pm}$8 mm | 1.078 | 0.36 | 64.164 |
| 18C16P $b_{pm}$3 mm | 0.511 | 9.72 | 70.926 |
| 18C16P $b_{pm}$4 mm | 0.658 | 6.46 | 68.554 |
| 18C16P $b_{pm}$5 mm | 0.763 | 3.04 | 63.569 |
| 18C16P $b_{pm}$6 mm | 0.866 | 0.66 | 60.149 |
| 18C16P $b_{pm}$7 mm | 0.915 | 3.12 | 54.457 |
| 18C20P $b_{pm}$2 mm | 0.450 | 0.17 | 75.034 |
| 18C20P $b_{pm}$3 mm | 0.649 | 0.11 | 72.138 |
| 18C20P $b_{pm}$4 mm | 0.820 | 0.12 | 68.368 |
| 18C20P $b_{pm}$5 mm | 0.917 | 0.06 | 61.153 |
| 18C20P $b_{pm}$6 mm | 1.004 | 0.08 | 55.767 |

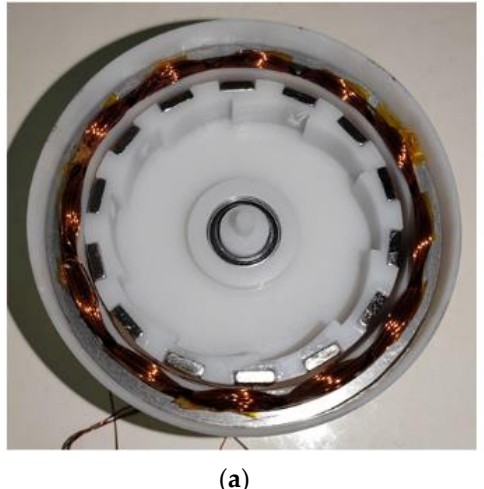 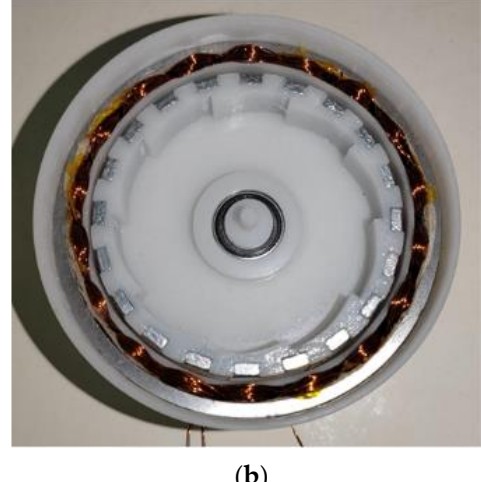

(**a**) (**b**)

**Figure 13.** Prototypes with different configurations: (**a**) 12C-14P with $b_{pm}$ = 6 mm; (**b**) 18C-20P with $b_{pm}$ = 4 mm.

The experimental results of the three-phase voltage waveforms with different configurations are shown in Figure 14. All voltage waveforms and their amplitudes correspond with the FEM results. After optimization, the voltage amplitude increases from the initial 0.5 V to about 1.0 V, and the PM mass slightly increases to 5.1 g. Furthermore, the harmonic components of the voltage waveforms are reduced, and exhibit a favorable fundamental sine.

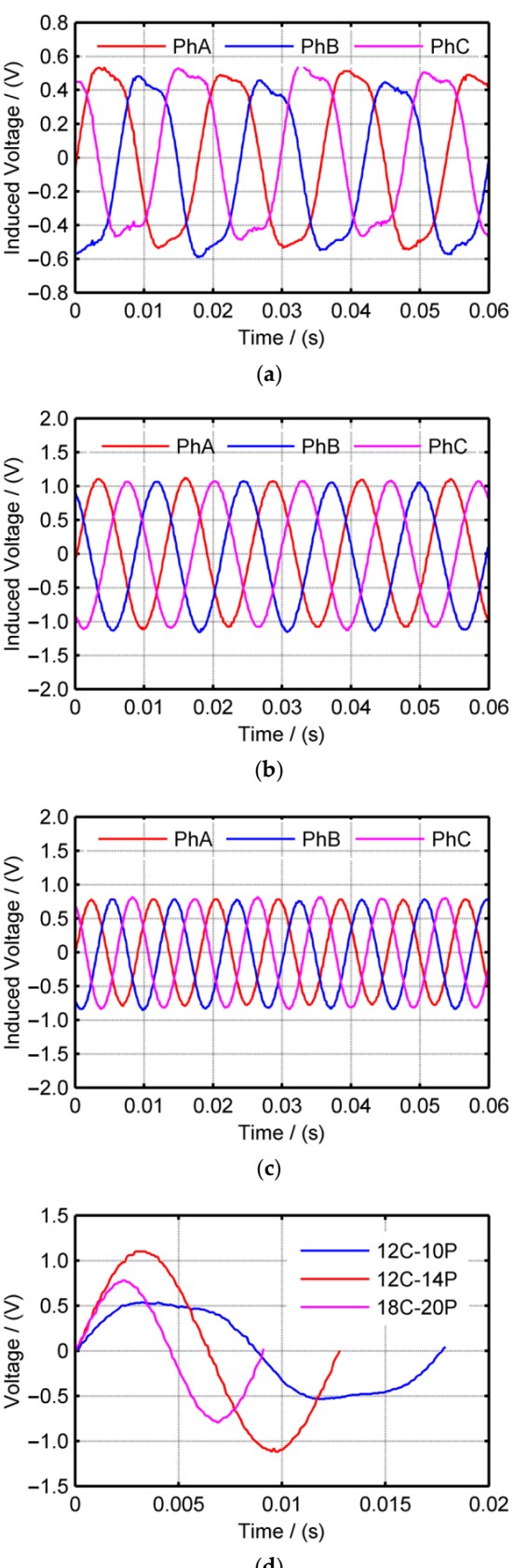

**Figure 14.** Experiment results of three-phase voltage waveforms at 600 rpm: (**a**) 12C-10P with $b_{pm}$ = 5 mm (initial); (**b**) 12C-14P with $b_{pm}$ = 6 mm (optimal); (**c**) 18C-20P with $b_{pm}$ = 4 mm; (**d**) Result comparison in one period of Phase A.

## 4. Experiment under Swing Motion

### 4.1. Test Bench for Swing Motion

With the optimal electromagnetic performance, the EEH prototypes were excited by different kinds of vibrations to validate their vibration energy-harvesting performance. The experimental setup for swing motion exciting is shown in Figure 15, where the swing motion is provided by a crank-rocker mechanism driven by a BLDC motor. Different conditions of swing excitation were supplied. The swing frequency of the crank-rocker mechanism was regulated by controlling the rotation speed of the motor (0~3000 rpm). Additionally, the swing angle can be changed by varying the length of the crank in a range from 20° to 80°. The length of the rocker, which is the distance from the pivot point to the center of the EEH, can be set to 132 mm, 102 mm or 72 mm. The electric outputs of the EEH prototype were displayed and stored through an oscilloscope.

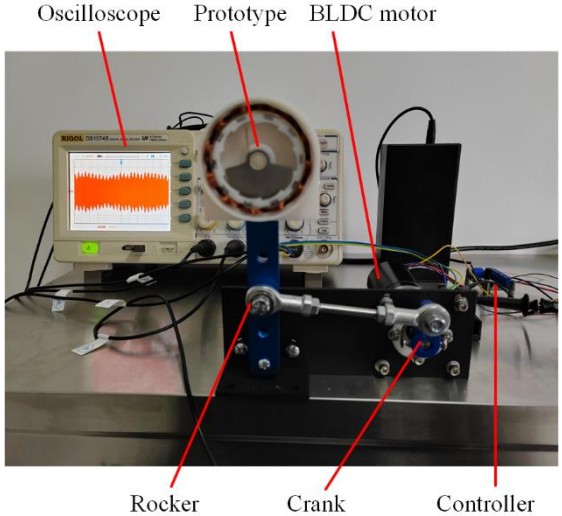

**Figure 15.** Experiment device for swing excitation.

### 4.2. Performance under Swing Motion

The length of the rocker was set to 132 mm, with a swing angle of 60° and a swing frequency of 2.5 Hz. Excited by the same swing motion, the open-circuit voltage outputs of the three manufactured prototypes are shown in Figure 16. Continuous voltage waveforms were generated. The optimal 12C-14P prototype produced the maximum amplitude voltage. The swing frequency was varied to explore the harvesting response performance, from 1.6 Hz to 3.5 Hz. Then, the swing angle was changed to 40°, and the frequency variation was conducted again.

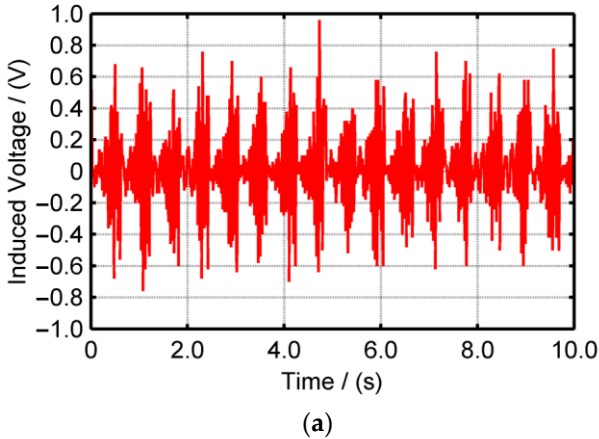

(**a**)

**Figure 16.** *Cont.*

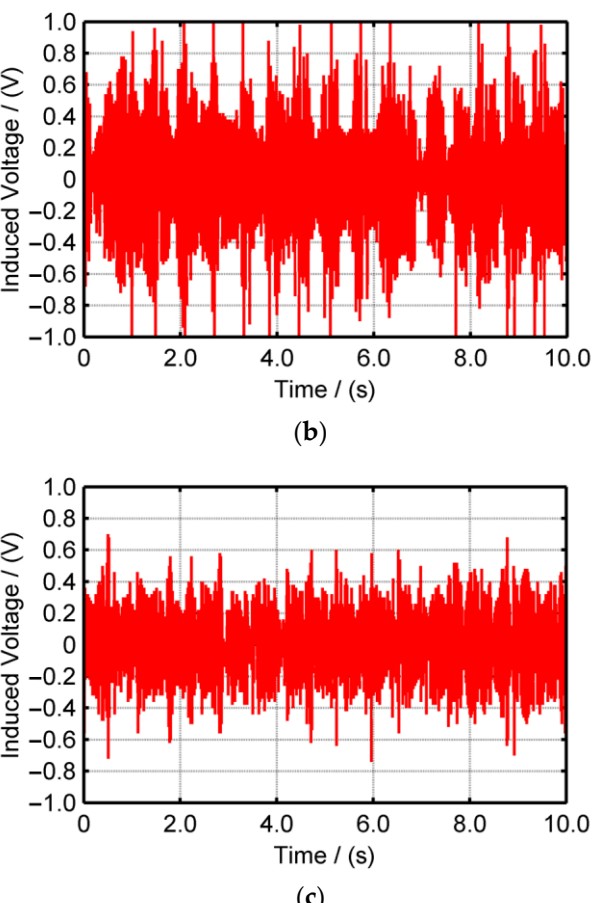

**Figure 16.** Output voltage of the three prototypes under swings with 2.5 Hz, 60°: (**a**) 12C-10P with $b_{pm}$ = 5 mm (initial); (**b**) 12C-14P with $b_{pm}$ = 6 mm (optimal); (**c**) 18C-20P with $b_{pm}$ = 4 mm.

The approximate values of the voltage amplitude under different conditions were compared, as shown in Figure 17. Generally, the voltage amplitude is enhanced with the swing angle and frequency, particularly when the frequency shifts between 2.5 Hz and 3.2 Hz. Under all swing conditions, the optimal 12C-14P prototype gained the highest amplitude when compared to the other two prototypes. This is because, when under the same rotor speed excited by the swing motion, the optimal 12C-14P prototype induces the maximum voltage amplitude. Moreover, with the increase of the swing frequency, the eccentric mass would exhibit a continuous circular motion that is opposite to the rotor direction, which cannot contribute to increasing the rotor speed. Therefore, the voltage amplitude trend reaches the peak value.

As shown in Figure 15, the swing vibration is provided by a crank-rocker mechanism. When the rocker swings at a low frequency, the harvesting energy is small if the weight of the eccentric mass is low. With the increased weight of the eccentric mass, the inertia and momentum increase, and the harvesting energy is enhanced. The initial eccentric mass with three copper blocks was 20 g. To detect the influence of the weight of the eccentric mass, a whole steel block of eccentric mass was fabricated, the weight of which was 60 g. After equipping the steel eccentric mass to the 12C-14P prototype, the swing motions with a 60° angle and different frequencies were conducted in sequence. It can be reasonably assumed that the whole steel structure can output a higher voltage amplitude. Furthermore, the significant characteristic is that an ultra-low frequency swing can be harvested with the higher inertia eccentric mass, as shown in Figure 18. Under the 1 Hz swing excitation, the initial eccentric mass hardly transfers to the rotor motion, and thus the continuous output voltage cannot be generated. In contrast, the whole steel eccentric mass can capture the ultra-low frequency swing, and then the speed of the rotor can be gradually enhanced to

reach stability. Moreover, after the swing is removed, the output voltage can persist over 12 s, which represents a long duration of the swing excitation. Therefore, the higher inertia eccentric mass should be considered under ultra-low, unstable or complicated vibrations.

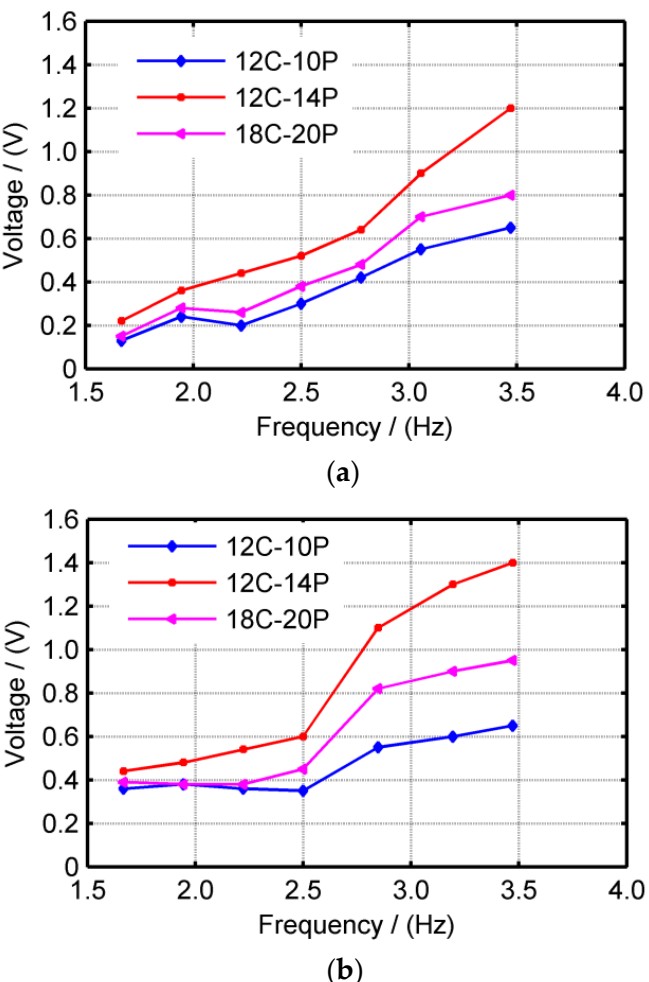

**Figure 17.** Frequency response of output voltage under different swing angles: (**a**) Swing 40°; (**b**) Swing 60°.

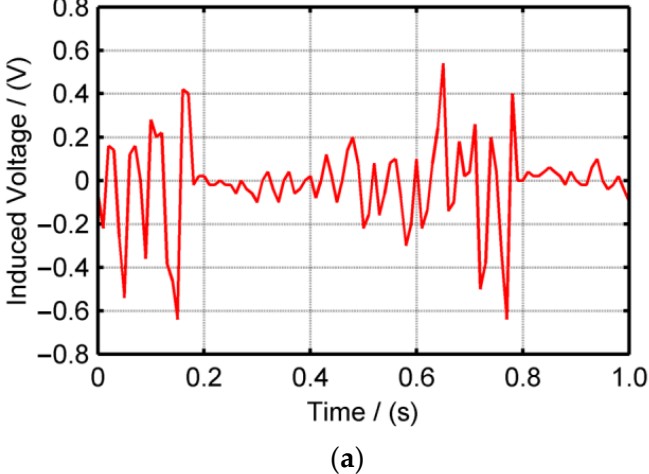

**Figure 18.** *Cont.*

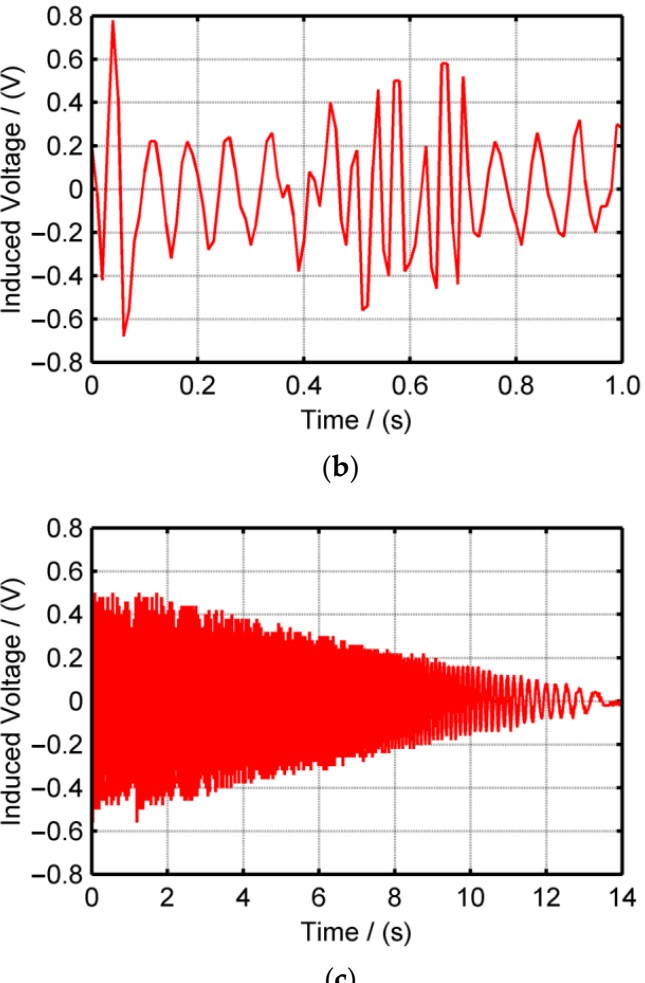

**Figure 18.** Output voltage of the 12C-14P prototype with two eccentric mass structures in ultra-low swing frequency (1Hz): (**a**) With initial structure under swing; (**b**) With whole steel block structure under swing; (**c**) With whole steel block structure after swing remove.

## 5. Experiment under Linear Reciprocation

### 5.1. Test Bench for Linear Reciprocation

The experimental setup for linear reciprocation exciting is shown in Figure 19, where the EEH is excited by a crank-slider mechanism driven by a BLDC motor. Different conditions of linear vibration were supplied. The motion frequency can be regulated by controlling the speed of the motor (0~3000 rpm), and the reciprocating stroke can be tuned by adjusting the effective length of the crank from 20 mm to 60 mm.

### 5.2. Performance under Linear Reciprocation

In the test, the length of the crank was set to 40 mm and was unchanged, and several frequencies from 2.2 Hz to 3.3 Hz were conducted. The output voltage amplitudes of different prototypes are shown in Figure 20. With the increase of frequency, the amplitudes increased in all three prototypes. The optimal 12C-14P prototype exhibits a larger amplitude than the others, especially over 2.7 Hz. However, when the frequency is higher than 3.0 Hz, the voltage amplitude does not increase. This is mainly because the eccentric mass in the EEH has a complex irregular motion characterized by the random shift between the swing and the multi-circle rotation.

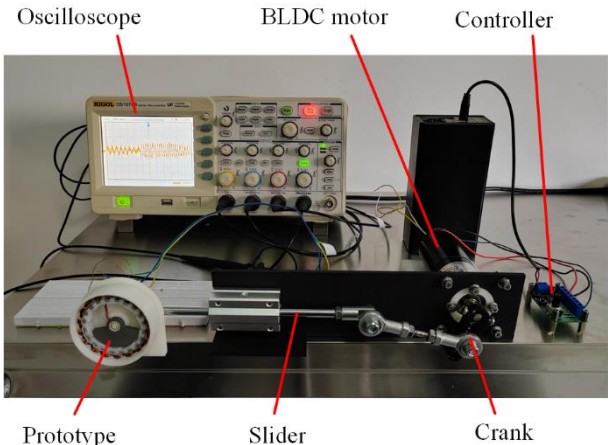

**Figure 19.** Experiment device for linear reciprocation.

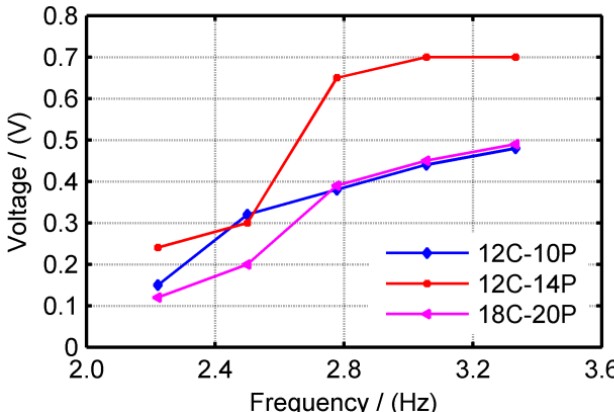

**Figure 20.** Frequency response of output voltage under 50 mm linear motion.

Furthermore, to indicate the effectiveness of the proposed optimization, the original device, presented in [25], which has been compared with a representative swing-based energy harvester and demonstrated the superiority, was tested under the same linear reciprocation experiment. As shown in Figure 21, the optimal prototype can induce larger volage amplitude and more continuous voltage waveform.

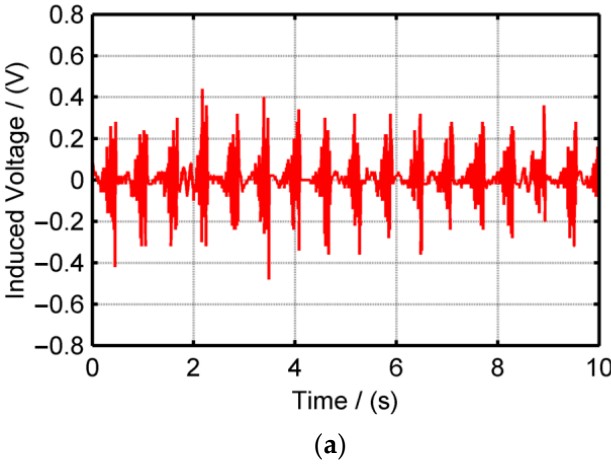

(**a**)

**Figure 21.** *Cont.*

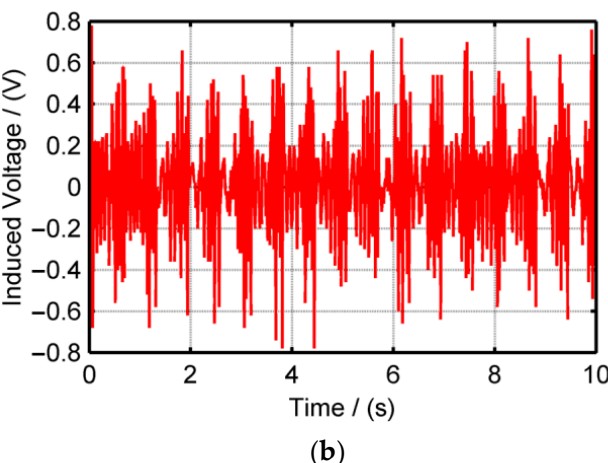

**(b)**

**Figure 21.** Output voltage of the original and optimal prototypes under linear reciprocation with 2.5 Hz (**a**) With original prototype; (**b**) With optimal prototype.

## 6. Conclusions

Electromagnetic characteristic analysis was carried out to optimize the output performance of the proposed EEH. A three-phase fractional coil per magnet pole was the basis structure, and the configurations of coil and pole, including 12C-10P, 12C-14P, 18C-16P and 18C-20P, were the four available candidates. Through FEM, the voltage waveforms and their harmonic amplitude are compared. Meanwhile, the effectiveness of FEM analysis effectiveness was confirmed by the prototype experimental results. Then, the optimal configuration and the PM dimension were determined to be 12C-14P with $b_{pm}$ = 6 mm, which induce over voltage 1.0 V and uses fewer PMs. On this basis, the vibration experiments were conducted, including swing and linear motion. The optimal prototype always generated the highest amplitude of output voltage. The vibration frequency existed within a favorable range that did not exceed 3.0 Hz, otherwise the eccentric mass would undergo multi-circle rotation. Moreover, the influence of the weight of eccentric mass was analyzed. Higher inertia eccentric mass should be considered under ultra-low, unstable or complicated vibrations. This EEH can be applied to exploit the ambient swing and linear vibration energy. For instance, the human arm and leg motion and the generated electric energy can be supplied to the wearable devices. However, low power applications and miniature devices are available. In further research, comparison with different types of EEH would be carried out for quantitative characteristic analysis.

**Author Contributions:** Conceptualization, J.L. and K.F.; methodology, J.L. and K.F.; software, C.Z.; validation, J.L. and C.Z.; formal analysis, J.L.; investigation, J.L.; resources, K.F.; data curation, J.L. and C.Z.; writing—original draft preparation, J.L.; writing—review and editing, K.F.; visualization, C.Z.; supervision, K.F.; project administration, K.F.; funding acquisition, J.L. All authors have read and agreed to the published version of the manuscript.

**Funding:** This research work is financially supported by the Project funded by Natural Science Foundation of China (No. 51605363), the Project supported by Natural Science Basic Research Plan in Shaanxi Province of China (No. 2021JM-124), and the Fundamental Research Funds for the Central Universities (No. XJS210403).

**Data Availability Statement:** The datasets used and analyzed during the current study are available from the corresponding author on reasonable request.

**Conflicts of Interest:** The authors declared no potential conflict of interest with respect to the research, authorship and publication of this article.

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
