# Peer review of "Performance Optimization of Ultralow-Frequency Electromagnetic Energy Harvester Driven by Eccentric mass"

_machines, doi:10.3390/machines11070743_

Round 1

Reviewer 1 Report

Ref. No.: Manuscript ID: machines -2414709

Title: Performance optimization of ultralow-frequency electromagnetic energy harvester driven by eccentric mass

In this paper, the authors present an enhancement of their studies about performance of an ultralow-frequency electromagnetic energy harvester driven by eccentric mass. I recommend publication once the comments below are addressed:

1) There is no reference to papers published in Machines Journal. If the authors think that their findings should be present in this forum (journal) then it is expected that papers on the topic of energy harvester published in the journal be referenced.

2) In Introduction Section there is no explicit research gap. The authors present some results from other researchers but the authors don’t state a question or a problem that has not been answered by any of those previous studies, which motivated the current research.

3) Several figures need better resolution, mainly those from Matlab.

4) There are problems with Eq. (2).

5) Avoid using a dot to express multiplication (see Eq. 3-5).

6) Details of the FEM modeling are not presented.

7) The authors must highlight the practical applications and limitations of the studied device.

Minor editing of English language required

Author Response

Response to reviewer

Reviewer 1

Comments and Suggestions for Authors

In this paper, the authors present an enhancement of their studies about performance of an ultralow-frequency electromagnetic energy harvester driven by eccentric mass. I recommend publication once the comments below are addressed:

1) There is no reference to papers published in Machines Journal. If the authors think that their findings should be present in this forum (journal) then it is expected that papers on the topic of energy harvester published in the journal be referenced.

Response: several papers about energy harvesting published in this journal have been cited.  

2) In Introduction Section there is no explicit research gap. The authors present some results from other researchers but the authors don’t state a question or a problem that has not been answered by any of those previous studies, which motivated the current research.

Response: The drawback of the conventional EEH has been indicated in the introduction section, large vibration amplitude is required, low efficiency and low electric power output. In order to capture low & ultralow frequency vibrations efficiently, an innovative eccentric mass combined with a two-layered cantilevered plectrum to capture low & ultralow frequency vibrations has been proposed in our previous study. However, the waveform and the efficiency of the electromagnetic induction effect did not be considered and optimized. Therefore, analysis and optimization design of the electromagnetic performance is carried out to obtain high efficiency, high power density, and low harmonic components of the inducted voltage.  

3) Several figures need better resolution, mainly those from Matlab.

Response: All the figures have been improved the resolution, especially the Matlab output figure.

4) There are problems with Eq. (2).

Response: The correct equation has been added in the revision Eq. (2), it is a mistake copying from the draft to the manuscript template. 

5) Avoid using a dot to express multiplication (see Eq. 3-5).

Response: All the equations have been deleted the dot symbol for multiplication.

6) Details of the FEM modeling are not presented.

Response: the FEM model and analysis process have been presented in the electromagnetic analysis section of the revision.

7) The authors must highlight the practical applications and limitations of the studied device.

Response: The practical applications and limitations of the studied device have been stated in the conclusion section.

Comments on the Quality of English Language

Minor editing of English language required

Response: The language spelling and grammar have been checked by the author, and revised by the language editing service.  

Reviewer 2 Report

The authors have investigated four different configurations of coil and pole for electromagnetic energy harvesters in an effort to find the best configuration. The topic is of interest to the research community. The prototype is adequately described, and the overall merit of this work is high. However, there are some comments that need to be addressed before final acceptance of the paper.

1. The title of the paper implies optimization of the performance. However, what has really been done is choosing the best performance out of only four configurations, which does not qualify as an optimization. So, please modify according to the main aim of the paper.

2. Equation (2) seems to be missing, please modify.

3. Please elaborate the governing equations of the FEM that the opensource software FEMM uses to solve the problem. What are the limitations and the assumptions for your simulation?

A moderate English Language copy-editing is required. Here and there, there are some grammatical and spelling mistakes. Please revise.

Author Response

Response to reviewer

Reviewer 2

Comments and Suggestions for Authors

The authors have investigated four different configurations of coil and pole for electromagnetic energy harvesters in an effort to find the best configuration. The topic is of interest to the research community. The prototype is adequately described, and the overall merit of this work is high. However, there are some comments that need to be addressed before final acceptance of the paper. 

  1. The title of the paper implies optimization of the performance. However, what has really been done is choosing the best performance out of only four configurations, which does not qualify as an optimization. So, please modify according to the main aim of the paper.

Response: In the presented EEH, three-phase stator and PM surface-mount rotor are employed. For this topology, the number of coils Q must be the multiple of 3 (three phase), and an even number can avoid the magnetic unbalance. So that Q can be set to 6, 12 or 18, etc., When Q less than 12, the leakage flux would become more, and when Q more than 18 coils the structure would become complicated. So that the preferable number of Q is 12 or 18. Correspondingly, the number of PM poles p should be an even number for the alternant polarity. Furthermore, higher winding factor can induce larger voltage amplitude. According to the Equations (1) to (5), four coil-pair combinations:

12C/10P (kw = 0.933), 12C/14P (kw = 0.933), 18C/16P (kw = 0.945), 18C/20P (kw = 0.945), exist higher winding factor. 

Therefore, it can be seen that the optimal configuration can be obtained among these four coil-pair combinations. And then, the key dimensions of the PM are optimized through electromagnetic analysis, and the yoke structure is also improved. So, the optimization has been completed.

  1. Equation (2) seems to be missing, please modify.

Response: The correct equation has been added in the revision Eq. (2), it is a mistake copying from the draft to the manuscript template. 

  1. Please elaborate the governing equations of the FEM that the opensource software FEMM uses to solve the problem. What are the limitations and the assumptions for your simulation?

Response: The FEM model calculating equations have been presented in the revision. The limitations of the FEM analysis is that the EEH model only can be excited by constant speed rotation, and no external disturbance is assumed. 

Comments on the Quality of English Language

A moderate English Language copy-editing is required. Here and there, there are some grammatical and spelling mistakes. Please revise.

Response: The language spelling and grammar have been checked by the author, and revised by the language editing service. 

Reviewer 3 Report

This gives a presentation of their small size voltage generators using electromagnetic sensing of angular or linear type motions. As such it would appear to be nicely eventually publishable. However

1. All abbreviations need definition, Some may be clear, as PM, but not so for EMDR.  2. All mathematical symbols need definition. Among those not clear ate m for (2)?, z in (3), h_pm and b_pm for the cantilever for which a figure would help. 3. To express k_m of (5) in terms of C & P. 4. To indicate what is the "yoke". 5.What does the "two" mean in Fig. 17? 6. Concerning the flexible portion of the cantilever: as it undergoes a lot of bending, how long will it last? 7. The programs/codes should be made available to a reader on request so a statement is needed how to get them. Note also that there is lots of data presented so the statement that it is "not applicable" should not be the case.

The English could use editing by a technical language editor as there are a number of different types of difficulties. Among others are at lines 77, 146, 178, 248.

Author Response

Response to reviewer

Reviewer 3

Comments and Suggestions for Authors

This gives a presentation of their small size voltage generators using electromagnetic sensing of angular or linear type motions. As such it would appear to be nicely eventually publishable.

However 

  1. All abbreviations need definition, Some may be clear, as PM, but not so for EMDR.

Response:  All abbreviations for their definitions have been confirmed.

  1. All mathematical symbols need definition. Among those not clear ate m for (2)?, z in (3), h_pm and b_pm for the cantilever for which a figure would help.

Response: All mathematical symbols have been checked for their definition. The correct equation has been added in the revision Eq. (2), it is a mistake copying from the draft to the manuscript template. hpm means the thickness of the PM, and bpm means the width of the PM, which have been indicated in Fig. 4.  

  1. To express k_m of (5) in terms of C & P.

Response: kd is the winding distribution factor, kc is the short pitch factor, and kw is the winding factor. The winding Factor kw is defined as the product of the winding distribution factor (kd) and the short pitch factor (kc). The distribution factor measured the resultant voltage of the distributed winding regards concentrate winding, and the short pitch factor is the measure of the number of armature slots between the two sides of a coil.

  1. To indicate what is the "yoke".

Response: The “yoke” is described in the revision. The yoke is a magnetic frame between tooth which can pass the magnetic flux, and also houses the coil windings.  

5.What does the "two" mean in Fig. 17?  

Response:  The “two” means the 12C-14P prototype with two different eccentric mass structures, one is the initial structure as shown in Fig. 4, the fan-shaped resin frame embedded with three identical cylindrical copper blocks; the other one is the whole steel eccentric mass.

  1. Concerning the flexible portion of the cantilever: as it undergoes a lot of bending, how long will it last?

Response: As shown in Fig. 2, when the eccentric mass is excited to spin clockwise, the cantilevered plectrum exhibits low stiffness and low frictional resistance to the rotor. The left elastic layer of the plectrum will be bended by the ratchet. If the clockwise motion maintains, the bending would continue. Once the eccentric mass is excited to spin anticlockwise, the bending also exists by the force between the ratchet and the rigid layer, but the amplitude of bending would diminish. In summary, bending of the elastic layer is sustained when the excitation of the eccentric mass exists. 

  1. The programs/codes should be made available to a reader on request so a statement is needed how to get them. Note also that there is lots of data presented so the statement that it is "not applicable" should not be the case.

Response: The statement has been added in the revision, the datasets used and analyzed during the current study are available from the corresponding author on reasonable request.

Comments on the Quality of English Language

The English could use editing by a technical language editor as there are a number of different types of difficulties. Among others are at lines 77, 146, 178, 248.  

Response: The language spelling and grammar have been checked by the author, and revised by the language editing service. 

Reviewer 4 Report

This paper proposes an electromagnetic energy harvester configuration. The comments are the following:

1) The most major comment is the fact that the paper lacks comparison to corresponding devices; thus the novelty part is significantly weak.

2) The authors should explain how the design of the proposed EEH is initiated. If it is based on a previous design, a proper reference should be included.

3) Equation 2 is just a simple determination of a value?

4) The comparison of the FEM result with the experimental one should be conducted in the same graph in Figure 5.

5) A comparison between the different coil-pole configurations must be conducted with respect to the harmonic generation.

6) Again, the experimental results of Figure 13 must be compared to the simulations in the same graph.

7) In Figure 15, the graphs should be placed in the same figure (with an appropriate resolution and window for time) in order to comprehend the differences in the peak voltage.

8) The authors mention in line 315 that the initial eccentric mass hardly transfers to the rotor motion. Is it due to the lower swing frequency (1 Hz)? Please comment on this.

9) The experimental analysis for the linear reciprocation is very weak. The authors should either improve or remove it.

The language manipulation is ok

Author Response

Response to reviewer

Reviewer 4

Comments and Suggestions for Authors

This paper proposes an electromagnetic energy harvester configuration. The comments are the following:

1) The most major comment is the fact that the paper lacks comparison to corresponding devices; thus the novelty part is significantly weak.

Response: The drawback of the conventional EEH has been indicated in the introduction section, large vibration amplitude is required, low efficiency and low electric power output. In order to capture low & ultralow frequency vibrations efficiently, an innovative eccentric mass combined with a two-layered cantilevered plectrum to capture low & ultralow frequency vibrations has been proposed in our previous study. However, the waveform and the efficiency of the electromagnetic induction effect did not be considered and optimized. Therefore, analysis and optimization design of the electromagnetic performance is carried out to obtain high efficiency, high power density, and low harmonic components of the inducted voltage. 

2) The authors should explain how the design of the proposed EEH is initiated. If it is based on a previous design, a proper reference should be included.

Response: In the introduction section of the revision, the previous work has been introduced and referenced.

3) Equation 2 is just a simple determination of a value?

Response: The correct equation has been added in the revision Eq. (2), it is a mistake copying from the draft to the manuscript template. 

4) The comparison of the FEM result with the experimental one should be conducted in the same graph in Figure 5.

Response: Because the experimental result in Fig. 5 (b) is a print screen derived from the oscilloscope when the according experiment is carried out, which can indicate the actual result directly. So the two figures are retained. To compare the FEM result and experimental result clearly, another figure has been added, including one phase one period voltage waveforms of the FEM and experimental results to be compared.  

5) A comparison between the different coil-pole configurations must be conducted with respect to the harmonic generation.

Response:  The comparison between the different coil-pole configurations have been added in Table. 1. 

6) Again, the experimental results of Figure 13 must be compared to the simulations in the same graph.

Response: The Figures 13 (a), (b), (c) are oscilloscope graphs for different prototype experiments. To compare the waveforms with different prototypes, another figure. 13 (d) has been added, including one phase one period voltage waveforms of the experimental results.

7) In Figure 15, the graphs should be placed in the same figure (with an appropriate resolution and window for time) in order to comprehend the differences in the peak voltage.

Response: In Figure 15, the oscilloscope graphs for different prototypes can indicate the whole trend and voltage density, and the peak voltage can be read and compared directly.

8) The authors mention in line 315 that the initial eccentric mass hardly transfers to the rotor motion. Is it due to the lower swing frequency (1 Hz)? Please comment on this.

Response: As shown in Fig. 14, the swing vibration is provided by a crank-rocker mechanism. If the weight of the eccentric mass is low, when the rocker swings at the low frequency (1Hz), the vibration energy can be captured is small. With the increase of the weight, the swing amplitude of the eccentric mass would enhance, and the captured energy would increase.

9) The experimental analysis for the linear reciprocation is very weak. The authors should either improve or remove it.

Response: The experimental analysis for the linear reciprocation is indicated that the proposed EEH can capture the linear type vibration energy. The operating condition is just the same as the above swing vibration. If necessary, some experiment about linear motion parameters can be conducted.

Comments on the Quality of English Language

The language manipulation is ok.   

Response: The language spelling and grammar have been checked by the author, and revised by the language editing service. 

Round 2

Reviewer 2 Report

No more comments.

Author Response

Author's Reply to the Review Report 

For Reviewer 2

Comments and Suggestions for Authors

No more comments.

Response:  The manuscript has been checked and revised carefully again, thank you very much! 

Reviewer 4 Report

The authors conducted a low effort to respond to the recommendations of the review. Specifically,

1) The print screen graphs from the oscilloscope are far from an ideal result presentation. The authors should extract the data and make the corresponding plots. For example, figure 6 (of the revised version) provides very good information since it validates the design. Additionally, Figure 14d is a very good representation of the data from Figures 14a-c, which can be removed since they are not easy to comprehend. The same for Figure 16a-c.

2) The authors included the Table that corresponds to the harmonic generation for different coil-pole configurations, but there is not any reference or comment in the manuscript.

3) A direct comparison to equivalent devices must be conducted not just a reference in the introduction. For example, some quantitative results in order to prove the superiority and the novelty of the proposed device.

The language manipulation is good.

Author Response

Author's Reply to the Review Report 

For Reviewer 4

Comments and Suggestions for Authors

The authors conducted a low effort to respond to the recommendations of the review. Specifically,

1) The print screen graphs from the oscilloscope are far from an ideal result presentation. The authors should extract the data and make the corresponding plots. For example, figure 6 (of the revised version) provides very good information since it validates the design. Additionally, Figure 14d is a very good representation of the data from Figures 14a-c, which can be removed since they are not easy to comprehend. The same for Figure 16a-c.

Response:  The experimental results have been extracted the data and made the corresponding plots, instead of the print screen graphs from oscilloscope with low resolution ratio. Thank you very much for your suggestions.  

2) The authors included the Table that corresponds to the harmonic generation for different coil-pole configurations, but there is not any reference or comment in the manuscript.

Response: I am very sorry for my mistake, the harmonic analysis among different coil-pole configurations has been included in the revision.

3) A direct comparison to equivalent devices must be conducted not just a reference in the introduction. For example, some quantitative results in order to prove the superiority and the novelty of the proposed device.

Response:  A direct comparison has been conducted in the revision. to indicate the effectiveness of the proposed optimization. The original device presented in [25], which has been demonstrated the superiority in our previous study, was compared under the same linear reciprocation experiment, as shown in Fig. 21. It can be seen that the optimal prototype can induce larger volage amplitude and more continuous voltage waveform.

Comments on the Quality of English Language

The language manipulation is good.

Response:  The language spelling and grammar have been checked by the author, thank you very much!

Round 3

Reviewer 4 Report

The authors put much more effort into this revision and the manuscript has been significantly improved in terms of presentation. However, the comparison is conducted only with the previous prototype of the same team. As commented in the previous revision a quantitative comparison (not a qualitative) should be conducted with other equivalent devices. This means in practice that at least 3 or 4 other works should be compared with the proposed device. Indeed, there are plenty of results in references 15-24 of this paper. Consequently, comparisons to these devices are required to clarify that this device is advantageous

Author Response

Author's Reply to the Review Report 

For Reviewer 4

Comments and Suggestions for Authors

The authors put much more effort into this revision and the manuscript has been significantly improved in terms of presentation. However, the comparison is conducted only with the previous prototype of the same team. As commented in the previous revision a quantitative comparison (not a qualitative) should be conducted with other equivalent devices. This means in practice that at least 3 or 4 other works should be compared with the proposed device. Indeed, there are plenty of results in references 15-24 of this paper. Consequently, comparisons to these devices are required to clarify that this device is advantageous 

Response: Thank you very much for your suggestions. The electromagnetic energy harvester (EEH) introduced in reference 15-18 require a relatively large vibration amplitude to make these harvesters operation operate efficiently. In reference 19-24, with the energy capture principle, the eccentric masses or pendulums are excited to swing at small angle and low speed, so these harvesters generally produce low electric power and voltage. In our previous research reference 25, the proposed EEH with eccentric mass can transform ultralow frequency vibrations and swings to uni-directional and continuous rotation with comparatively high speeds. The presented original prototype has been compared with a representative swing-based energy harvester, the comparison results demonstrate its feasibility and superiority in capturing ultralow-frequency vibration energy. On this basis, the proposed optimization method in this manuscript can improve the induced voltage effectively. On the other hand, the manufacturing of the prototypes may require a period of time. The quantitative comparison with other energy harvesting devices is a favorable method for characteristic analysis, so it will be conducted in our further research.